# In-Situ Combination of Bipolar Membrane Electrodialysis with Monovalent Selective Anion-Exchange Membrane for the Valorization of Mixed Salts into Relatively High-Purity Monoprotic and Diprotic Acids

**DOI:** 10.3390/membranes10060135

**Published:** 2020-06-26

**Authors:** Haiyang Yan, Wei Li, Yongming Zhou, Muhammad Irfan, Yaoming Wang, Chenxiao Jiang, Tongwen Xu

**Affiliations:** 1CAS Key Laboratory of Soft Matter Chemistry, Collaborative Innovation Center of Chemistry for Energy Materials, School of Chemistry and Material Science, University of Science and Technology of China, Hefei 230026, China; oceanyan@ustc.edu.cn (H.Y.); ymzhou@mail.ustc.edu.cn (Y.Z.); engr_muhammad.irfan@hotmail.com (M.I.); jcx11@ustc.edu.cn (C.J.); 2Hefei ChemJoy Polymer Materials, Co., Ltd., Hefei 230601, China; liwei8991@126.com

**Keywords:** bipolar membrane, electrodialysis, monovalent selective anion-exchange membranes, zero liquid discharge, coal mining industry

## Abstract

The crystalized mixed salts from the zero liquid discharge process are a hazardous threat to the environment. In this study, we developed a novel electrodialysis (SBMED) method by assembling the monovalent selective anion-exchange membrane (MSAEM) into the bipolar membrane electrodialysis (BMED) stack. By taking the advantages of water splitting in the bipolar membrane and high perm-selectivity of MSAEM for the Cl^−^ ions against the SO_4_^2−^ ions, this combination allows the concurrent separation of Cl^−^/SO_4_^2−^ and conversion of mixed salts into relatively high-purity monoprotic and diprotic acids. The current density has a significant impact on the acid purity. Both the monoprotic and diprotic acid purities were higher than 80% at a low current density of 10 mA/cm^2^. The purities of the monoprotic acids decreased with an increase in the current density, indicating that the perm-selectivity of MSAEM decreases with increasing current density. An increase in the ratio of monovalent to divalent anions in the feed was beneficial to increase the purity of monoprotic acids. High-purity monoprotic acids in the range of 93.9–96.1% were obtained using this novel SBMED stack for treating simulated seawater. Therefore, it is feasible for SBMED to valorize the mixed salts into relatively high-purity monoprotic and diprotic acids in one step.

## 1. Introduction

High-salinity wastewater is usually considered for wastewater with total dissolved solids up to 10,000 ppm. To date, large volumes of high-salinity wastewater have been discharged from various industrial sections, such as the coal mining industry, petroleum and petrochemical industry, pulp and paper industry, power generation industry and fertilizer industry, among others. The composition of high-salinity wastewater is usually very complex and the wastewater contains various organic components, monovalent and multivalent inorganic salts such as Cl^−^, SO_4_^2−^, Na^+^ and Ca^2+^, among others. The inappropriate discharge of high-salinity wastewater will result in deleterious consequences such as the destruction of ecosystems, environmental contamination and salinization of soil [1,2,3]. The high-salinity wastewater treatment system directly affects the sustainability of modern industrial societies.

Considering the coal mining industry, the Chinese government encourages the development of modern coal chemical technologies based on coal gasification technologies and other advanced chemical technologies to produce oil, syngas, olefins and ethylene glycol owing to the special energy resource structure of “rich coal, less gas, lack of oil” in China [4]. Notably, coal mining is an intensive water consumption process. It is estimated that 0.5 tons of water are consumed to produce 1–2 tons of coal. However, the coal resources and water resources in China are in reverse distribution. Approximately 70% of coal mines are located in water-scarce areas and about 40% experience severe water shortage problems [5]. Conversely, coal mining produces approximately 3–6 billion tons of wastewater per year. The coal mining industry faces a dilemma: the discharge of large volumes of wastewater and the severe water scarcity problem. The strict regulations on wastewater disposal and lack of water availability are driving the implementation of a zero liquid discharge (ZLD) system [6,7]. Figure 1 indicates a typical wastewater treating procedure for the ZLD process, including biological treatment systems such as the traditional aerobic activated sludge process, aerobic granular sludge and the sequencing batch reaction (SBR) system, ultrafiltration system (i.e., UF, to remove the organic matters), nanofiltration system (i.e., NF, to separate the mono- and divalent salts) system, reverse osmosis system (i.e., RO, to concentrate the brine) system, membrane concentrating system such as electrodialysis (ED) and disc-tube reverse osmosis (DTRO) as well as the evaporation system. In recent years, the ZLD concept is developing rapidly since the legislation of the “Water Pollution Prevention and Control Action Plan” (alternatively known as the “Water Ten Plan”) in 2015. Theoretically, the implementation of a ZLD system can augment the water supply, eliminate the disposal of waste streams and valorize the valuable by-products. However, in reality, the ZLD process faces two crucial challenges during practical operation. On one hand, the ZLD system has a high capital cost and intensive energy input. All the soluble salts are finally required by thermal evaporation to generate the solid salts. It is a huge energy-consuming process even though emerging thermal evaporation technologies such as mechanical vapor recompression (MVR) and multiple-effect evaporation are developing rapidly. On the other hand, the evaporated and crystalized salts are usually mixtures of monovalent and divalent salts as well as organic matters that are difficult to reuse. Table 1 shows the estimated waste salt discharging capacities from the coal chemical industry in China. It can be seen that large amounts of waste salts are generated every year, which will become a hazardous threat to the environment. If the evaporated salts are not appropriately treated, the ZLD will become meaningless from the original aspiration of resource recycling. Therefore, how to valorize the mixed salts from the ZLD process has become the bottleneck for the ZLD process. 

Currently, the chlor-alkali industry is the largest salt-consuming industry. The principle of chlor-alkali is a membrane electrolysis process that converts NaCl into chlorine, hydrogen and sodium hydroxide in the presence of a direct electric current field. Unfortunately, there are stringent requirements for the salts in the chlor-alkali industry, but the recycled salts from the ZLD process cannot meet these requirements. In contrast, bipolar membrane electrodialysis (BMED) [9,10], which has less strict requirements for the feed compared with membrane electrolysis, is a perfect solution for the valorization of recycled salts into acids and bases in the ZLD process. 

BMED is an advanced electro-membrane process that combines the water splitting in the bipolar membrane and directional migration of ionic components in the conventional electrodialysis process [11,12]. The saline wastewater inputted to the BMED system can be effectively converted to acids and bases, achieving a closed loop industrial production. As a consequence, there is no need to evaporate the saline effluents and there will be no disposal of waste salts. Numerous studies have proved the feasibility of BMED for the production of acids and bases from industrial saline wastewater [13,14,15,16]. However, the saline wastewater usually contains both monovalent and divalent anions, but the conventional BMED can only convert the mixed salts into mixed acids such as HCl and H_2_SO_4_, and bases. Owing to the low values of mixed acids, the technological sustainability and viability of BMED is restricted during the valorization of salts into acids and bases in the ZLD process. To address this problem, selectrodialysis (SED), a novel electrodialysis assembly system, was firstly proposed by Zhang et al. [17] in 2012. The authors assembled monovalent selective ion exchange membranes into the conventional ED stack to achieve the separation of sulphate from a NaCl/Na_2_SO_4_ mixture. In a subsequent study [18], the authors further developed an integrated SED-struvite reactor to improve phosphate recovery efficiency using the high perm-selectivity of MSAEM for Cl^−^ ions against H_x_PO_4_^y-^ ions. Similarly, Xu et al. [19] investigated SED for the selective removal of arsenic and monovalent ions from a brackish water reverse osmosis concentrate. Cohena and co-authors [20] used SED for reducing chloride and sodium while preserving most of the hardness ions in the groundwater for irrigation. Zhang et al. [21] performed SED for the separation of divalent ions from a seawater concentrate to enhance the purity of coarse salt. After that, Reig et al. [22] ex situ combined SED with the BMED process to achieve monovalent–divalent ions separation and acids and bases production. However, the experiments were performed in two steps. In the first step, SED was used for the separation of Cl^−^ and SO_4_^2−^. In the second step, BMED was used for the conversion of NaCl and Na_2_SO_4_ into their corresponding acids (HCl and H_2_SO_4_) and base (NaOH). Recently, Qiu et al. [23] also ex situ integrated SED with BMED for the production of lithium hydroxide from salt lakes. This combination was also performed in two stages. Li^+^ ions were separated from Li^+^/Mg^2+^ mixtures by SED in the first stage, then Li^+^ ions were converted to LiOH by BMED in the second stage. Aspired from the advantages of the monovalent selective ion exchange membrane and BMED, in our previous study [24], we in situ integrated SED with the mono-selective cation-exchange membrane and BMED for selectively regenerating monovalent ions and producing acids/bases in a single route. In this way, the pretreatment procedure to remove the multivalent cations, which cause membrane fouling, can be eliminated. 

Apart from the concentrated brine from the RO plant, the wastewater from industrial societies such as the coal mining industry is rich in monovalent and divalent anions, and the valorization of these mixed salts is indispensable to the sustainability of the ZLD process. Herein, we in situ integrated BMED with MSAEM in a novel electrodialysis stack (SBMED). The MSAEM was placed between the bipolar membrane and the conventional anion-exchange membrane. By taking advantage of the perm-selectivity of MSAE for Cl^−^ ions against the SO_4_^2−^ ions, it is possible to valorize the mixed salts into high-purity monoprotic and diprotic acids in one step. Therefore, the main objectives of this study are: (i) to test the feasibility of the in situ combination of SED and BMED for the production of monoprotic and diprotic acids; and (ii) to test the effects of the current density and mixed salt components on the purities of monoprotic and diprotic acids. 

## 2. Experimental Section

### 2.1. Material

The membranes used in the experiments were Neosepta AMX (anion-exchange membrane, Tokuyama Co., Tokyo, Japan), Neosepta CMX (cation-exchange membrane, Tokuyama Co., Tokyo, Japan), ACS (monovalent selective anion-exchange membrane) and Neosepta BP-1E (bipolar membrane, Tokuyama Co., Tokyo, Japan). Their properties are listed in Table 2. All other chemicals were of analytical grade and used as received. Deionized water was used throughout the experiment.

### 2.2. Setup

A schematic diagram of the SBMED is illustrated in Figure 2. Especially, this experimental setup was comprised of (1) a cathode and an anode, which were made of titanium coated with ruthenium; (2) alternatively arranged bipolar membrane, MSAEM, anion-exchange membrane, cation-exchange membrane and bipolar membrane; the effective area of each membrane was 20 cm^2^ and the spacer thickness was 10 mm; (3) two electrodes that were connected to a direct current power supply (WYL1703, Hangzhou Siling Electrical Instrument Ltd., Hangzhou, China); the electrode was purchased from Baoji Qixin Titanium Co. Ltd., Baoji, China; the voltage drop across the stack was directly read from the indicator on the power supply; (4) five circulation compartments, namely an acid I chamber, acid II chamber, salt chamber, base chamber and electrode chamber were formed; and (5) peristaltic pumps (BT-100 L, Baoding Lead Fluid Technology Co., Ltd., Baoding, China) which were used to circulate the solutions with a flow rate of 200 mL/min. The conductivity was measured by a DDBJ-350 conductivity meter (Shanghai Leici Equipment Co. Ltd., Shanghai, China).

Three kinds of mixed salt aqueous solutions, including 0.05 mol/L NaCl + 0.05 mol/L Na_2_SO_4_, 0.06 mol/L NaCl + 0.04 mol/L Na_2_SO_4_ and 0.08 mol/L NaCl + 0.02 mol/L Na_2_SO_4_, with the volume of 1 L, were fed into the salt compartment. To increase the initial solution conductivity, the acid I compartment and base compartment were fed with 0.05 mol/L HCl and 0.05 mol/L NaOH, with each volume of 200 mL, respectively. The acid II compartment was filled with 200 mL of deionized water. The anode and cathode compartments were circulated together with a 0.3 mol/L Na_2_SO_4_ (200 mL) as the rinse solution. The SBMED experiments were conducted at the galvanostatic mode with the current density in the range of 10–50 mA/cm^2^ and the voltages were directly recorded from the board of the DC power supply.

### 2.3. Operating Principle of the SBMED

Different from the conventional BMED process, an MSAEM was specially placed between the bipolar membrane and the anion-exchange membrane as illustrated in Figure 2. In the presence of an electric current field, the water splitting in the bipolar membrane would be excited, generating a large amount of H^+^ and OH^−^ ions. In the salt compartment, both the Cl^−^ and SO_4_^2−^ ions would be transported toward the anode direction under the driving force of the electric current field and then migrate through the anion-exchange membrane. Due to the existence of the MSAEM, the Cl^−^ ions could be further transported across the MSAEM into the acid I compartment and combined with the protons generated by the bipolar membrane, while the SO_4_^2−^ ions were retarded by the MSAEM and would be kept in the acid II compartment. Since most anion-exchange membranes have poor blockage against protons [25,26,27], to maintain the electro-neutrality in the acid II compartment, it was easy for the protons to migrate though the MSAEM into the acid II compartment in the presence of a direct current field. Meanwhile, the Na^+^ ions would be transported through the CEM into the base compartment and combined with OH^−^ ions generated by the bipolar membrane. Therefore, the mixed salt could be converted into relatively high-purity monoprotic acid in the acid I compartment and diprotic acid in the acid II compartment. 

### 2.4. Analyses and Calculations

The Cl^−^ and SO_4_^2−^ concentrations were determined by inductively coupling a plasma optical emission spectrometer (ICP-OES) (Optima 7300 DV, USA). The acid and base concentrations were titrated using phenolphthalein as an indicator.

Theoretically, the acid I and acid II compartments were rich in monoprotic acid and diprotic acid after the experiment, respectively. Therefore, the purity of monoprotic acid ($P_{Cl_{}^{-}}$) in the acid I compartment and diprotic acid ($P_{so_{4}^{2-}}$) in the acid II compartment were calculated using Equations (1) and (2), respectively.
(1)$$P_{Cl^{-}}=\frac{C_{Cl^{-}}}{C_{Cl^{-}\;}+C_{SO_{4}^{2-}}}\times 100\%$$
(2)$$P_{so_{4}^{2-}}=\frac{C_{SO_{4}^{2-}}}{C_{Cl^{-}\;}+C_{SO_{4}^{2-}}}\times 100\%$$
where $C_{Cl^{-}\;}$ and $C_{SO_{4}^{2-}}$ are the molar concentrations of chloride and sulfate ions in the acid I or acid II compartment, respectively. 

The energy consumptions to produce HCl (*E_HCl_*) and NaOH (*E_NaOH_*) were calculated using Equations (3) and (4), respectively.
(3)$$E_{HCl}=\int_{0}^{t}\frac{UIdt}{(C_{t,HCl}-C_{0,HCl})\;V_{a}M_{HCl}}$$
(4)$$E_{NaOH}=\int_{0}^{t}\frac{UIdt}{(C_{t,NaOH\;}-C_{0,NaOH})\;V_{b}M_{NaOH}}$$
where *U* (V) is the voltage drop across the SBMED stack; *I* (A) is the current applied; *C_t_*_,*HCl*_ and *C*_0,*HCl*_ (mol/L) are the HCl concentration in the acid I compartment at time t and time 0, respectively; *V_a_* and *V_b_* (L) are the volume of the acid I compartment and base compartment, respectively; and *M_HCl_* and *M_NaOH_* are the molar masses of HCl and NaOH, respectively. 

The current efficiency in the acid I compartment (*η_Acid_*_,*I*_) and base compartment (*η_Base_*) of the SBMED process was calculated using Equations (5) and (6), respectively.
(5)$$\eta_{Acid,\;I}=\frac{Z\left(C_{t,HCl}-C_{0,HCl}\right)V_{a}F}{NIt}\times 100\%$$
(6)$$\eta_{Base}=\frac{Z\left(C_{t,NaOH}-C_{0,NaOH}\right)V_{b}F}{NIt}\times 100\%$$
where *C_t_*_,*HCl*_ and *C*_0,*HCl*_ (mol/L) are the HCl concentration in the acid I compartment at time t and time 0, respectively; *V_a_* and *V_b_* (L) are the volume of the acid I compartment and base compartment, respectively; *Z* is the ion’s absolute valence (*Z* = 1 for HCl and NaOH); *F* is the Faraday constant (96,485 C/mol); *I* (A) is the current applied; and *n* is the repeating unit number (*n* = 1) in this study. 

## 3. Results and Discussion

### 3.1. Effect of Current Density 

The SBMED was an electric-driven membrane separation process, and thus the current density was a dominant factor affecting the electrodialysis performance. Therefore, typical current densities in the range of 10–50 mA/cm^2^ were chosen for the experiment. Figure 3 indicates the effect of the current density on the voltage drop across the SBMED stack and the conductivity change in the salt compartment during the experiment. Generally, the limiting current density of the commercial bipolar membrane was very low [28,29] (up to 1 mA/cm^2^ for the modern bipolar membrane). The current in the bipolar membrane was carried by the diffusion of salt ions when the current was lower than the limiting current density, while the current was mainly carried by water dissociation products when the limiting current density was exceeded. When the current density applied was higher than the limiting current density, the water molecules would be dissociated into H^+^ and OH^−^ ions at the interphase of the bipolar membrane. Therefore, the voltage drops and conductivities decreased as a function of time as shown in Figure 3, indicating the introduction of H^+^ and OH^−^ ions to the SBMED stack. However, the pattern of conductivity curves was quite different from the voltage drop change curves. The conductivities were almost linearly decreased as a function of time, indicating the steady depletion of salt during the experiments. This is because the conductivity of the salt compartment in the SBMED was operated under a galvanostatic mode. Thus, the amounts of ions transported out of the salt compartment were linearly dependent on the amount of Coulomb applied at the stack. The conductivities in the acid and base compartments increased quickly at the early stage of the experiments (0–10 min) due to the occurrence of water dissociation in the bipolar membrane. The salt conductivities continuously decreased and this was caused by the migration of salts out from the salt compartment. Therefore, the overall voltage drop became stable at the latter stage of the experiments. 

Figure 4 indicates the effect of the current density on the evolutions of the acid and base concentrations. It could be seen that both the acid and base concentrations increased linearly as a function of time, indicating the steady water dissociation in the bipolar membrane. The greater current provided the higher acid and base concentrations. It is a logical truth that the water dissociation is accelerated with increasing the current density, owing to the second Wien effect [30,31]. Another possible reason is that the autoprotolysis effect of the membrane functional groups and water was enhanced with the increasing current density. The base concentrations were a little higher at the same moment compared with the acid concentrations. This phenomenon was likely attributable to several causes. On one hand, there were two acid compartments in comparison with one base compartment. The dissociated H^+^ ions firstly entered into the acid I compartment, and a small fraction of H^+^ ions then further transported into the acid II compartment. The migration of H^+^ ions from the acid I compartment into the acid II compartment was exacerbated owing to two aspects. On one hand, the H^+^ ions would be migrated toward the cathode direction in the presence of an electric current field. It is not difficult for the H^+^ ions to be transported through the MSAEM. On the other hand, the H^+^ ions were migrated due to the electroneutrality requirement for the stream solution in the acid II compartment. Theoretically, if the perm-selectivity of the MSAEM was 100%, the SO_4_^2−^ ions migrated from the salt compartment could not enter into the acid I compartment and would be retarded in the acid II compartment. To maintain the solution electroneutrality, it was favorable for the migration of H^+^ ions through the MSAEM under the influence of the chemical potential. Secondly, the selectivity of the anion-exchange membrane against the protonic co-ions was lower than the selectivity of the cation-exchange membrane against the anionic co-ions [27,32,33]. Due to the small ionic size and low valence of the proton, most commercial anion-exchange membranes have poor blockage performance against H^+^ ions. In addition, the selectivity of the membrane decreased with an increase in the current density [34]. Therefore, the concentration gradient between the acid and base was accelerated at a higher current density, as shown in Figure 4. Nevertheless, the mixed salts can be converted into high-value acids and bases as expected. 

Figure 5a shows the effect of the current density on acid purity in the acid I compartment. The purity of HCl would be 100% if the perm-selectivity of chloride ions against sulfate ions was 100%. However, here, the HCl purities decreased gradually as a function of time. For instance, the HCl purities are in the range of 92.3–94.5% at the running time of 20 min, then the HCl purities decreased to the range of 65.1–82.0% at the running time of 120 min. The decrease in acid purity was caused by the decreased perm-selectivity of MSAEM as the running time was prolonged. The decreasing tendency of the monovalent selectivity of MSAEM over time was consistent with the reported studies as follows. Cohen et al. [20] found that a major decline in monovalent selectivity was observed during treating groundwater for irrigations. They also explained the loss of monovalent selectivity was possibly caused by membrane fouling. Zhang et al. [17] calculated the perm-selectivity of a commercial MSAEM for the separation of sulfate from a NaCl/Na_2_SO_4_ mixture; they found the perm-selectivity of PC-MVA was close to 1 from the beginning until four hours and declined to 0.85 in the fifth hour. Zhao et al. [34] investigated the perm-selectivity of different membranes and found that the value of perm-selectivity decreased as a function of time. All the above studies suggested that the transport of co-ions highly affected the perm-selectivity as the running time was prolonged. Notably, it was also found that the purities of monoprotic acid decreased with an increase in the current density. The perm-selectivity of MSAEM for monovalent ions against divalent ions can be significantly improved when the current density applied was lowered. For a low current density of 10 mA/cm^2^, both the monoprotic and diprotic acid purities were higher than 80% after the experiment, as shown in Figure 5b. The fluxes of both Cl^−^ and SO_4_^2−^ ions through the MSAEM were accelerated at a higher current density. With respect to a higher current density, the flux of the divalent ions was more pronouncedly affected compared with the flux of the monovalent ions. Thus, more SO_4_^2−^ ions were transported into the acid I compartment, reducing the purity of the target monoprotic acids. In fact, the decrease in the monovalent perm-selectivity of the ion exchange membrane with increasing current density was consistent with numerous studies [35,36,37,38,39]. The reason should be referred to the structure of the monovalent ion exchange membrane. To achieve the selectivity of monovalent ions against multi-valent ions with the same charge, three strategies are usually adopted [40]. The first commonly used strategy is to introduce a tight oppositely charged layer on the surface of the general ion exchange membrane. Due to the electrostatic repulsion effect, the formation of an additional layer resulted in a higher repulsion for the multi-valent counterions than for the monovalent counterions. The second adopted strategy is to control the pore size of the membranes by increasing the cross-linking degree of the ion exchange membrane. Due to the pore sieving effect, the larger size multivalent counterions were less permeated through the membrane compared with the smaller-size monovalent counterions. The third used strategy is to adjust the hydrophilicity of membranes by introducing a polyelectrolyte layer on the surface of the membrane. Due to the specific interaction effect between the polyelectrolyte layer and the counterions, the monovalent counterions with the lower Gibbs hydration energy were easier to be transported through the membrane compared with the multivalent counterions. In our case, the ion transport rate through the membrane was determined by the ion mobility in the membrane, which was significantly affected by the current field. When the current density increased, the driving force became stronger and this led to a suppression of the electrostatic interaction and steric hindrance effects. Moreover, according to the mathematical models developed by Zabolotsky et al. [41,42], the perm-selectivity of monovalent ions against divalent ions would approach to a stable value that was independent of the membrane properties and the space charge region when the current density applied was close to or higher than the limiting current density. This means that the perm-selectivity of the membrane at a high current density was controlled solely by the diffusivity of the ions in the solution. Therefore, the perm-selectivity of the membrane decreased at a high current density. It should be noted that the co-ion exclusion leakage also affected the perm-selectivity of the membrane. The co-ion exclusion leakage was exacerbated at a high current density, so the transport number of anions decreased. However, considering the perm-selectivity of MSAEM was the transport number of the monovalent anions against that of the divalent anions, it was still unknown the transport number of which anions decreased much greater at a higher current density. Therefore, an elaborate measurement of the transport numbers of monovalent anions, divalent anions and cations with the dependence of a different current density is required. This will be carried out in our further studies. Nevertheless, this study reached a relatively high purity of monoprotic acid by an in situ combination of BMED with MSAEM. 

### 3.2. Effect of Monovalent and Divalent Salt Component 

A complex component of mixed salt wastewater was discharged from a variety of industries, such as the coal mining industry, the printing and dyeing industry, the power plant industry, the pulp and paper industry, among others. To further investigate the effect of the mixed salt component on the SBMED performance, three kinds of electrolyte solution with the same anionic strength, i.e., 0.05 mol/L NaCl + 0.05 mol/L Na_2_SO_4_, 0.06 mol/L NaCl + 0.04 mol/L Na_2_SO_4_ and 0.08 mol/L NaCl + 0.02 mol/L Na_2_SO_4_, were investigated. As shown in Figure 6, the conductivities in the salt compartment linearly decreased as a function of time. Similar to Figure 3, the voltage drops across the SBMED stack dramatically decreased at the initial stage of the experiment and then became stable during the later period of the experiment. The voltage drop curves and the conductivity curves proved the successful operating of the SBMED experiments. This means that the salt ions were migrated out of the salt compartment while the water splitting in the bipolar membrane brought the continuous inputting of H^+^ and OH^−^ ions into the acid compartment and base compartment, respectively. Notably, the limiting equivalent conductance of the SO_4_^2−^ ions in the bulk solution was 80 S·cm^−2^·equiv^−1^, that was slightly higher than Cl^−^ ions of 76 S·cm^−2^·equiv^−1^ [43]. Among the above three kinds of mixed salts with the same anionic strengths, the 0.05 mol/L Na_2_SO_4_ + 0.05 mol/L NaCl feed has the highest cationic strength. Therefore, the 0.05 mol/L Na_2_SO_4_ + 0.05 mol/L NaCl feed has the highest solution conductivity in the salt compartment among the three kinds of feed. In view of acid purity, it was clearly shown that acid purity decreased as a function of time: this is similar to the results in Figure 5 with the same explanations. The 0.08 mol/L Na_2_SO_4_ + 0.02 mol/L NaCl feed produced the highest acid purities in the range of 90.0–93.5%, while the 0.05 mol/L Na_2_SO_4_ + 0.05 mol/L NaCl feed produced the lowest acid purities in the range of 67.3–87.7%. With an increase in the molar ratio of the monovalent Cl^−^ to the divalent SO_4_^2−^ ions, there was an increased portion of monovalent ions to be transported through the MSAEM. Therefore, the monoprotic acid purity increased with an increase in the molar ratio of the monovalent ions to the divalent ions in the feed. 

Figure 7a demonstrated the current efficiency of three kinds of mixed salt components. The current efficiencies in the base compartment for three kinds of mixed salts were in the range of 86.0–92.5%, while the current efficiencies in the acid I compartment were in the range of 50.8–52.1%. Considering that the H^+^ ions generated from the bipolar membrane existed both in the acid I and acid II compartments, the current efficiencies in both the acid and base compartments were acceptable. The energy consumption for the produced monoprotic acid and base were calculated and presented in Figure 7b. The energy consumptions for the production of monoprotic acid were in the ranges of 6.84–9.10 kWh/kg HCl for three kinds of mixed salt components. The energy consumption for the production of HCl decreased with an increase in the molar ratio of the monovalent Cl^−^ to the divalent SO_4_^2−^ ions: this was due to an increase in the HCl purity in the acid I compartment with the increase in the molar ratio. In contrast, there was no obvious difference in the energy consumptions for the production of NaOH for the three kinds of mixed salt, which were in the range of 3.16–3.60 kWh/kg NaOH. These data were consistent with the current efficiencies in Figure 7a. It should be noted that the electrode rinse was also included in these energy consumption calculations. The energy consumption would become much smaller if the number of membrane pairs increased to the industrial level, such as several hundreds of pairs of membranes for a pair of electrodes. Notably, a specific advantage of our system consisting of bipolar membranes along with monovalent selective ion exchange membranes was the possibility for the generation of relatively high-purity monoprotic and diprotic acids, which have high added value and could be used for the regeneration of ion exchangers or as chemicals for the upstreaming processes. In contrast, the conventional bipolar membrane electrodialysis could only obtain mixed acids, which have low values and are only suitable for pH adjustment processes.

### 3.3. Simulated Seawater

The above section has proved that the increase in the molar ratio of the monovalent to divalent component was beneficial for the final monoprotic acid purity. For seawater or brackish water desalination, the recovery rate of reverse osmosis for seawater or brackish water desalination is usually around 60%: a large amount of concentrate RO brines are discharged without reuse. The composition of these brines is very similar to that of seawater, which contains both monovalent and divalent anions. To further broaden the applicability of this integrated method, we also tested the application in situ of the combination of bipolar membrane electrodialysis with the monovalent ion selective anion-exchange membrane for the valorization of simulated seawater. Therefore, a simulated seawater with a Cl^−^ concentration of 0.546 mol/L and SO_4_^2−^ concentration of 0.0281 mol/L was further investigated for the SBMED process. Figure 8 shows the SBMED performances with the simulated seawater. The voltage drop across the stack decreased as a function of time, indicating the smooth progress of the experiment. The acid and base concentrations linearly increased as a function of time. The final base and acid concentrations were around 0.30–0.35 mol/L after five hours of running. Notably, stable and relatively high purities of monoprotic acid in the range of 93.9–96.1% were obtained via this novel SBMED configuration. Compared with the conventional BMED stack that only obtains mixed acids, this SBMED was feasible to valorize the mixed salts into relatively high-purity monoprotic and diprotic acids in one step. The purities of the target monoprotic and diprotic acids were highly dependent on the perm-selectivity of the MSAEM. 

## 4. Conclusions

A novel electrodialysis was developed by assembling an MSAEM into the BMED stack. This novel cell configuration allowed the concurrent separation of Cl^−^/SO_4_^2−^ and conversion of mixed salts into relatively high-purity monoprotic and diprotic acids. The effects of the current density, molar ratio of monovalent to divalent anions on the voltage drop, acid/base concentration and acid purity have been systematic investigated. It was found that the monoprotic acid purities decreased with an increase in the current density. Both the monoprotic and diprotic acid purities were higher than 80% at a low current density of 10 mA/cm^2^. The molar ratio of monovalent to divalent anions has a significant influence on the SBMED performance. An increase in the molar ratio of monovalent to divalent anions in the feed was beneficial to increase the purity of the monoprotic acid. For three kinds of mixed salt, the current efficiencies in the base compartment were in the range of 86.0–92.5%, while the current efficiencies in the acid I compartment were in the range of 50.8–52.1%. Monoprotic acids with high-purity in the range of 93.9–96.1% were obtained via this novel SBMED stack using simulated seawater as the feed. In conclusion, it was feasible for SBMED to valorize the mixed salts into relatively high-purity monoprotic and diprotic acids in one step. The perm-selectivity of the commercial MSAEM needs to be improved to further enhance the purity of both monoprotic and diprotic acids. 

## Figures and Tables

**Figure 1 membranes-10-00135-f001:**
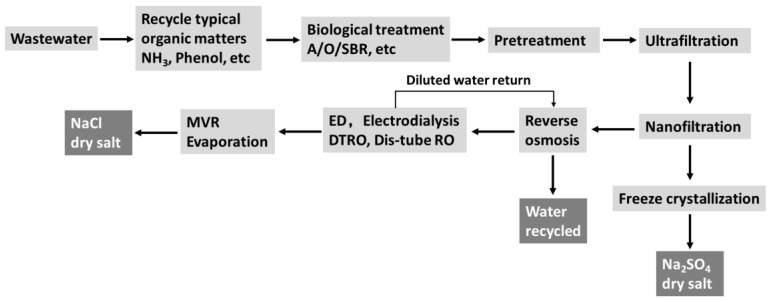
A typical zero liquid discharge wastewater treating procedure.

**Figure 2 membranes-10-00135-f002:**
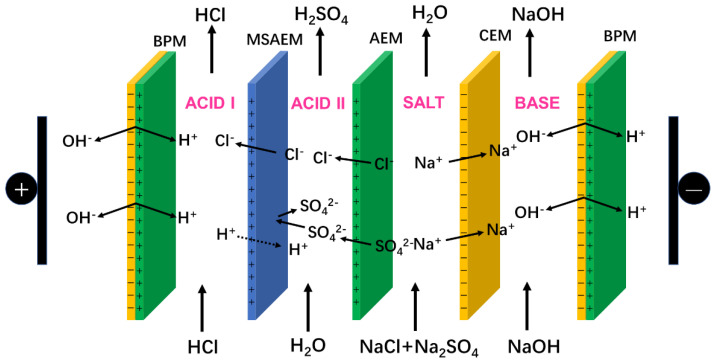
Schematic diagram for the novel electrodialysis stack (SBMED) process. BPM, bipolar membrane; AEM, anion-exchange membrane; CEM, cation-exchange membrane; MSAEM, mono-selective anion-exchange membrane.

**Figure 3 membranes-10-00135-f003:**
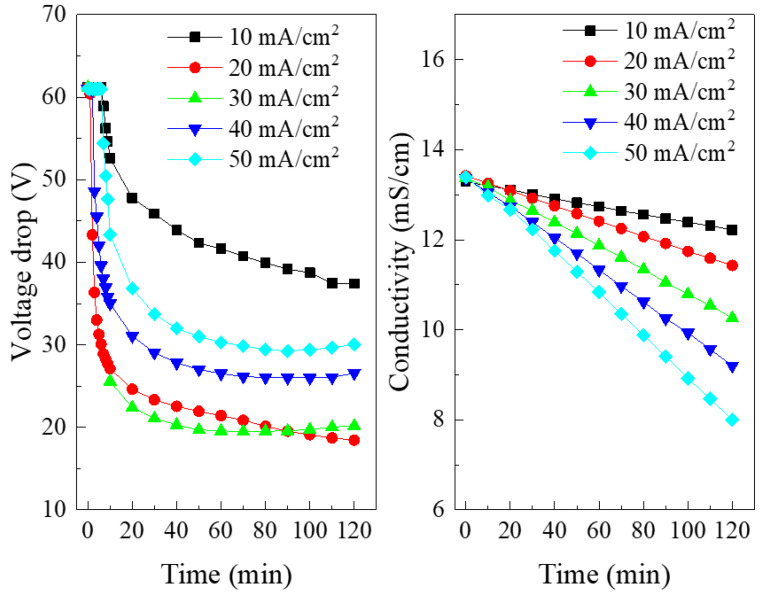
Effect of the current density on the voltage drop across the SEDBM stack and on the conductivity in the salt compartment.

**Figure 4 membranes-10-00135-f004:**
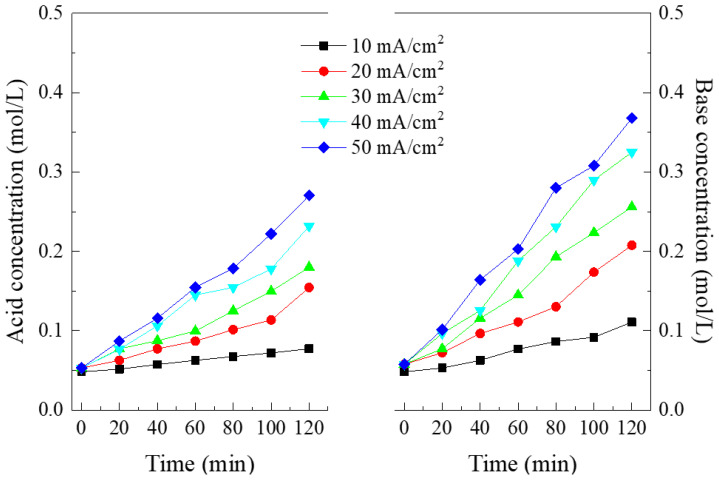
Effect of the current density on the evolution of the acid concentration in the acid I compartment and base concentration as a function of time.

**Figure 5 membranes-10-00135-f005:**
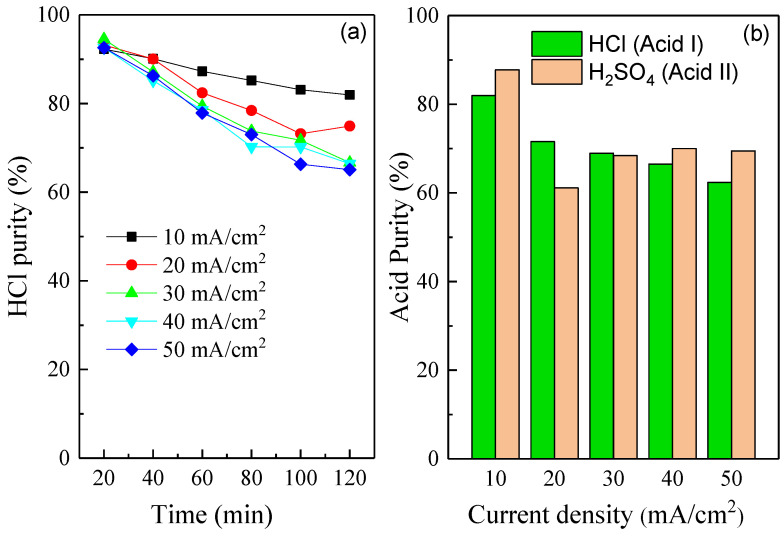
Effect of the current density on acid purity. (**a**) The evolution of HCl purity in the acid I compartment as a function of time; (**b**) the final HCl and H_2_SO_4_ purity in the acid I and acid II compartments after the experiments, respectively. Other experimental conditions: feed in salt compartment, 0.05 mol/L NaCl + 0.05 mol/L Na_2_SO_4_; electrode rinse solution, 0.3 mol/L Na_2_SO_4_; flow rate of each circulation, 200 mL/min.

**Figure 6 membranes-10-00135-f006:**
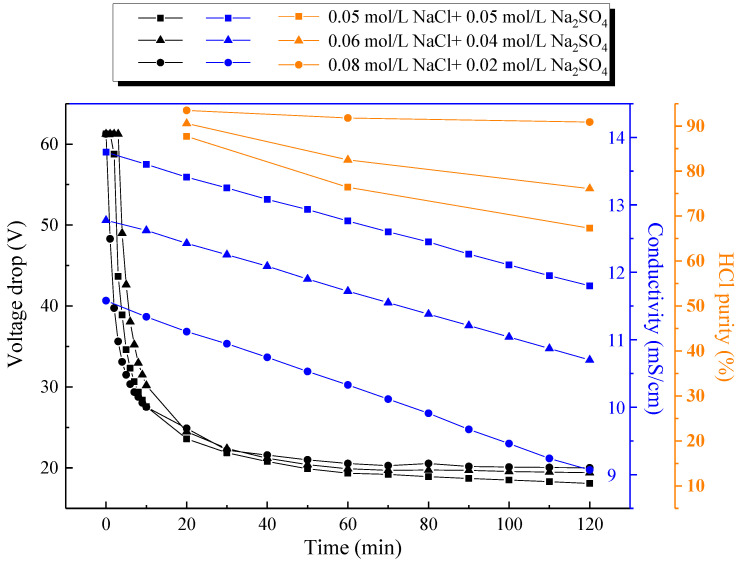
Effect of the monovalent and divalent components on the voltage drop across the SBMED stack, conductivity evolution curves in the salt compartment and HCl purity in the acid I compartment. Other experimental conditions: current density, 20 mA/cm^2^; electrode rinse solution, 0.3 mol/L Na_2_SO_4_; flow rate of each circulation, 200 mL/min.

**Figure 7 membranes-10-00135-f007:**
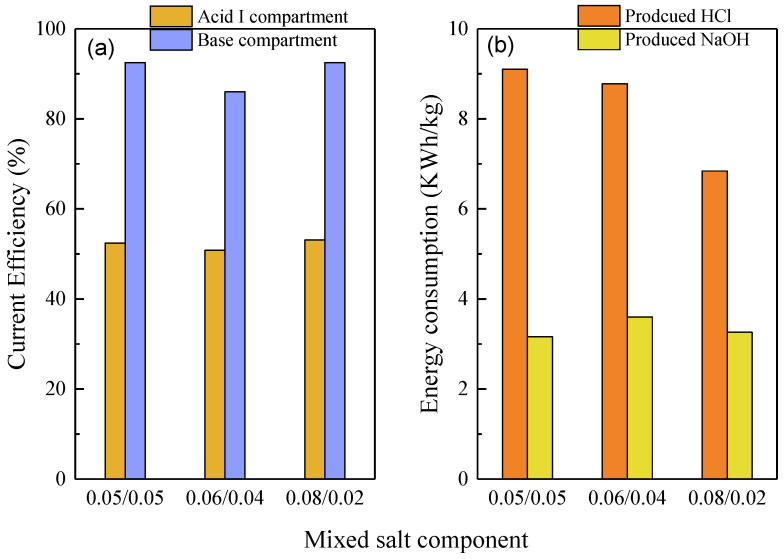
The influence of the mixed salt component on the current efficiency and energy consumption. (**a**) Current efficiency in the acid I and base compartment for the different salt components; (**b**) energy consumption for the produced HCl and NaOH.

**Figure 8 membranes-10-00135-f008:**
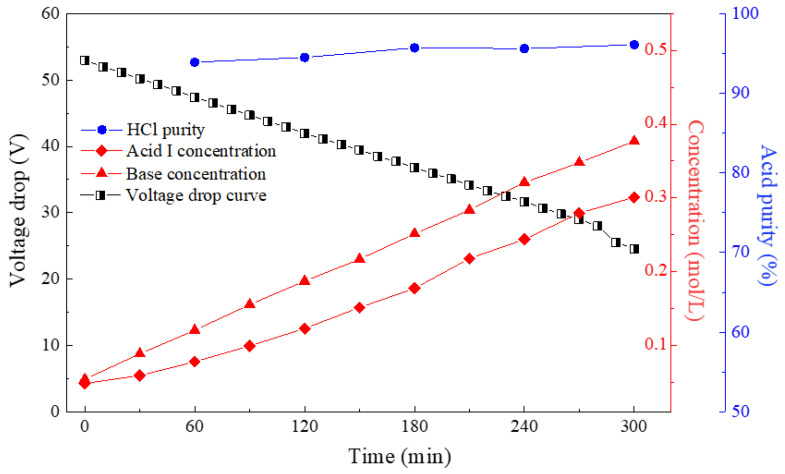
The SBMED performances, i.e., voltage drop curve as a function of time, HCl purity in the acid I compartment, acid concentration curve in the acid I compartment and base concentration curve in the base compartment using simulated seawater.

**Table 1 membranes-10-00135-t001:** Estimation of waste salt discharge capacity from the coal chemical industry in China ^a^.

Coal Transformation Type	Developed Scale	Waste Salt Discharge Capacity (10^4^ t)	Under Development Scale	Estimated Waste Salt Discharge Capacity (10^4^ t)
Coal to oil (10^4^ t/a)	640	64	800	80
Coal to syngas (10^8^ Nm^3^/a)	131	21.3	222	36
Coal to olefin (10^4^ t/a)	887	88.7	330	33
Coal to ethylene glycol (10^4^ t/a)	285	46.3	305	49.6
Total		220.3		198.7

^a^ The data were collected based on the book [8].

**Table 2 membranes-10-00135-t002:** The main properties of the membranes used for the experiments ^a.^

Membrane Type	Thickness (µm)	IEC (mmol/g)	Resistance (Ω·cm^2^)	Voltage Drop ^c^ (V)	Efficiency (%)
Neosepta BP-1E	220	-	-	1.2	>98
Neosepta CMX	170	1.5–1.8	3.0 ^b^	-	-
Neosepta AMX	140	1.4–1.7	2.4 ^b^	-	-
Neosepta ACS	130	1.4–2.0	3.8 ^b^	-	-

^a^ The data are collected from the product brochure provided by the company; ^b^ Equilibrated with a 0.5 mol/L NaCl solution at 30 °C; ^c^ measured with 1 mol/L NaOH and HCl at 100 mA/cm^2^ at 30 °C.

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
