# Peer review of "In-Situ Combination of Bipolar Membrane Electrodialysis with Monovalent Selective Anion-Exchange Membrane for the Valorization of Mixed Salts into Relatively High-Purity Monoprotic and Diprotic Acids"

_membranes, 2020, doi:10.3390/membranes10060135_

Round 1

Reviewer 1 Report

Investigation on bipolar membrane electrodialysis for the production of acids and bases is a long-prevailing topic which also currenty gaining attention from many broad perspectives with respect to the circular economy or power-to-x. The authors have performed a study in this context, investigating the impact of some parameters on system performance. The organization of the manuscript is ok, but some major issues need to be addressed before publication.

  1. Get help with improving the languages.
  2. Revise your title: The term “mono-selective” is not precise. Mono what?
  3. It’s not clear why the authors used simulated seawater while they claim the problem and challenges of wastewater in the introductory part. Justify or amend.
  4. Because most anion-exchange membranes have poor blockage
  5. l.175. “Because most anion-exchange membranes have poor blockage against the H+ ions, it was easy for the protons to migrate though the MSAEM into the Acid II compartment.” I don’t understand these statements – how can proton easily go through MSAEM?
  6. 4.l.216. “Generally, the limiting current density is lower than 5 mA/cm2 for a typical commercial bipolar membrane.” Modern commercial bipolar membranes even that the current is carried by water dissociation products even at lower current density (up to 1 mA/cm2). If you don’t do experiments in this range, you have to depict the behavior at low current ranges.
  7. 7.l.273. “It was a logically truth that the water dissociation is accelerated with increasing the current density owing to the Second Wien Effect”. Is there any effect of the autoprotolysis of the membrane functional grounds + water on enhancing water dissociation?  
  8. 9.l. 303. “This means that the perm-selectivity of membrane at high current density was controlled solely by the diffusivity of the ions in solution. Therefore, the perm-selectivity of the membrane decreased at high current density.” Is there a co-ion exclusion leakage at a high current density which migh limits the permselectivity - comment on this.
  9. 11.l.359. “The energy consumption would become much smaller if the number of membrane pairs increased to an industrial level, such as several hundred pairs of membranes for a pair of electrodes.” How?
  10. Add a short note on the economic aspect related to the specific advantage of your system consisting of bipolar membranes along with monovalent selective ion exchange membranes.

Author Response

Reviewer #1’s Comment: Investigation on bipolar membrane electrodialysis for the production of acids and bases is a long-prevailing topic which also currently gaining attention from many broad perspectives with respect to the circular economy or power-to-x. The authors have performed a study in this context, investigating the impact of some parameters on system performance. The organization of the manuscript is ok, but some major issues need to be addressed before publication.

Response. We sincerely appreciate the reviewer’s time and expertise in reviewing our manuscript. The reviewer’s constructive comments and suggestions help us improving the quality of the manuscript. We have studied your comments carefully and have made revisions. The point-to-point to the reviewer’s comments are listed as followings.

Q1. Get help with improving the languages.

A1. We have proofread the manuscript carefully to correct the grammatical, spelling and typo errors. In addition, we also asked several colleagues who are skilled speakers of English language to improve the English.

Q2. Revise your title: The term “mono-selective” is not precise. Mono what?

A2. Thanks for the comments. Now we have revised the term “mono-selective” as “Monovalent selective”.

Q3. It’s not clear why the authors used simulated seawater while they claim the problem and challenges of wastewater in the introductory part. Justify or amend.

A3. Thanks for the comments. In fact, the main background of this proposed in-situ combination of bipolar membrane with mono-valent ion selective anion exchange membrane is focused on wastewater. But we believe that the proposed method is also feasible for treating the other water that contain both the mono- and divalent anions. For seawater or brackish water desalination, the recovery rate of reverse osmosis for seawater or brackish water desalination is also around 60%; a large amounts of concentrate RO brines are discharged without reuse. The composition of these brines is very similar to that of seawater, which contains both monovalent and divalent anions. To further broaden the applicability of this integrated method, we also tested the application in-situ combination of bipolar membrane electrodialysis with mono-valent ion selective anion-exchange membrane for the valorization of simulated seawater. The reasons are justified in the revised manuscript in Line 390-396.

Q4. Because most anion-exchange membranes have poor blockage, l.175. “Because most anion-exchange membranes have poor blockage against the H+ ions, it was easy for the protons to migrate though the MSAEM into the Acid II compartment.” I don’t understand these statements – how can proton easily go through MSAEM?

A4. Thanks for the reviewer’s comments. The proton leakage against the anion exchange membrane is a common phenomenon in electrodialysis process, there are usually two mechanisms [1-3], one is the classical Grotthus mechanism in which the proton migrates from one water molecule to another and the other one is the classical co-ion leakage associated with the transport of the sorbed electrolyte. In our case, to maintain the electro-neutrality in Acid II, it was easy for the protons to migrate though the MSAEM into the Acid II compartment at the presence of a direct current field. To make clarity, this sentence has revised with the addition of references as “Because most anion-exchange membranes have poor blockage against the H+ ions [25-27], to maintain the electron-neutrality in the Acid II, it was easy for the protons to migrate though the MSAEM into the Acid II compartment at the presence of a direct current field”.

References:

[1] Lorrain, Y., Pourcelly, G., Gavach, C., 1996. Influence of cations on the proton leakage through anion-exchange membranes. J. Membr. Sci. 110, 181-190.

[2] Lorrain, Y., Pourcelly, G., Gavach, C., 1997. Transport mechanism of sulfuric acid through an anion exchange membrane. Desalination 109, 231-239.

[3]Beck, A., Ernst, M., 2015. Kinetic modeling and selectivity of anion exchange in Donnan dialysis. J. Membr. Sci. 479, 132-140.

Q5. 4.l.216. “Generally, the limiting current density is lower than 5 mA/cm2 for a typical commercial bipolar membrane.” Modern commercial bipolar membranes even that the current is carried by water dissociation products even at lower current density (up to 1 mA/cm2). If you don’t do experiments in this range, you have to depict the behavior at low current ranges.

A5. Thanks for the reviewer’s comment. We agree with the reviewer’s comments that the limiting current density for modern commercial bipolar membrane is very low (up to 1 mA/cm2). This sentence has revised as “Generally, the limiting current density of commercial bipolar membrane was very low (up to 1 mA/cm2 for modern bipolar membrane). The current was carried by the diffusion of salt ions when the current was lower than the limiting current density; while the current was mainly carried by water dissociation products when the limiting current density was exceeded”.

Q6. 7.l.273. “It was a logically truth that the water dissociation is accelerated with increasing the current density owing to the Second Wien Effect”. Is there any effect of the autoprotolysis of the membrane functional grounds + water on enhancing water dissociation?  

A6. Thanks for the comments. The autoprotolysis effect of membrane functional groups and water was enhanced with increasing in current density. This reason was added in the revised manuscript in Line 244-246.

Q7. 9.l. 303. “This means that the perm-selectivity of membrane at high current density was controlled solely by the diffusivity of the ions in solution. Therefore, the perm-selectivity of the membrane decreased at high current density.” Is there a co-ion exclusion leakage at a high current density which might limits the permselectivity - comment on this.

A7. Thanks for the reviewer’s very interesting comments. The co-ion exclusion leakage is exacerbated at a high current density, so the transport numbers of anions decrease. For this point, the co-ion exclusion leakage will affect the perm-selectivity of membrane. But considering the perm-selectivity of MSAEM was the transport number of monovalent anions against that of the divalent anions, it was still unknown the transport number of which anions decreased much greater at higher current density. Therefore, an elaborate measurement of the transport number of monovalent anions, divalent anions, cations with the dependence of different current density is required. This will be carried out in our further studies. This comment has added in the revised manuscript in Line 314-321.

Q8. 11.l.359. “The energy consumption would become much smaller if the number of membrane pairs increased to an industrial level, such as several hundred pairs of membranes for a pair of electrodes.” How?

A8. In this study, the voltage electrode is a few volts, and the electrode chamber consumed a considerable fractional of the total stack voltage. The energy consumption would be smaller if the voltage drop in the electrode chamber was neglected. If the number of membrane pairs increased to an industrial level, such as several hundred pairs of membranes for a pair of electrodes, the influence of electrode chamber could be neglected. Therefore, we stated that the energy consumption would become much smaller if the number of membrane pairs increased to an industrial level, such as several hundred pairs of membranes for a pair of electrodes.

Q9. Add a short note on the economic aspect related to the specific advantage of your system consisting of bipolar membranes along with monovalent selective ion exchange membranes.

A9. Thanks for the reviewer’s comments. Notably, a specific advantage of our system consisting of bipolar membranes along with monovalent selective ion exchange membranes was the possibility for the generation of relatively high purity monoprotic and diprotic acid products, which have high add-value and could be used for regeneration of ion exchanger or as chemicals for the upstreaming process. In contrast, the conventional bipolar membrane electrodialysis could only obtain mixed acids, which have low values and are only suitable for pH adjustment processes. This information has added in the revised manuscript in Line 376-382.

Reviewer 2 Report

The authors designed an interesting electrodialysis system by combining bipolar membrane electrodialysis with mono-selective anion-exchange membrane for the valorization of mixed salts wastewater.

This paper has the potential to be accepted, but some points have to be clarified or fixed before can be accepted.

  • Where did you get the electrode? It would be better if you provide the company information.
  • What is the function of the rinse solution in this system (line 158)? What is the consideration of using Na2SO4 as a rinse solution in here?
  • How to measure the conductivity (line 215)? Please provide sufficient information in the paper.
  • It would be better if the time on the base concentration chart (Figure 4) is in ascending order.
  • There was a typing error of "produce" in Figure 7. In my opinion, it's better to use "produced" or "product"?

Author Response

Reviewer #1’s Comment: The authors designed an interesting electrodialysis system by combining bipolar membrane electrodialysis with mono-selective anion-exchange membrane for the valorization of mixed salts wastewater.

This paper has the potential to be accepted, but some points have to be clarified or fixed before can be accepted.

Response. We appreciate the reviewer’s positive and constructive comments and suggestions on our manuscript. We have studied your comments carefully and have made revisions.

Q1. Where did you get the electrode? It would be better if you provide the company information.

A1. Thanks for the comment. The electrode was purchase from Baoji Qixin Titanium Co. Ltd., China. This information has added in the revised manuscript. This information has added in the revised manuscript in Line 149-150.

Q2. What is the function of the rinse solution in this system (line 158)? What is the consideration of using Na2SO4 as a rinse solution in here?

A2. The main function of rinse solution is worked as a mediate for the transport of ions between the anode and cathode and to generate a circular cycle for the ions transfer. The reason for the chosen of Na2SO4 as a rinse solution is that it is a cheap and very easy obtained chemical, and there are no emissions of notorious chlorines in electrode reactions during the experiment for this kind of electrolyte.  

Q3. How to measure the conductivity (line 215)? Please provide sufficient information in the paper.

A3. The conductivity was measured by a DDBJ-350 conductivity meter (Shanghai Leici Equipment Co. Ltd., China). This information has added in the revised manuscript in Line 154-155.

Q4. It would be better if the time on the base concentration chart (Figure 4) is in ascending order.

A4. Thanks for the reviewer’s kind suggestion. The time on base concentration chart has changed in the ascending order as suggested.

Q5. There was a typing error of "produce" in Figure 7. In my opinion, it's better to use "produced" or "product"?

A5. Thanks for the reviewer’s kind suggestion and we have corrected this typing errors.

Round 2

Reviewer 1 Report

The manuscript is improved.